# Broadband All-Optical THz Modulator Based on Bi_2_Te_3_/Si Heterostructure Driven by UV-Visible Light

**DOI:** 10.3390/mi14061237

**Published:** 2023-06-12

**Authors:** Yayan Xi, Yixuan Zhou, Xueqin Cao, Jing Wang, Zhen Lei, Chunhui Lu, Dan Wu, Mingjian Shi, Yuanyuan Huang, Xinlong Xu

**Affiliations:** Shaanxi Joint Laboratory of Graphene, State Key Laboratory of Photon-Technology in Western China Energy, International Collaborative Center on Photoelectric Technology and Nano Functional Materials, Institute of Photonics & Photon-Technology, School of Physics, Northwest University, Xi’an 710069, China

**Keywords:** THz modulator, Bi_2_Te_3_/Si heterostructure, modulation depth, photogenerated carrier

## Abstract

All-optical terahertz (THz) modulators have received tremendous attention due to their significant role in developing future sixth-generation technology and all-optical networks. Herein, the THz modulation performance of the Bi_2_Te_3_/Si heterostructure is investigated via THz time-domain spectroscopy under the control of continuous wave lasers at 532 nm and 405 nm. Broadband-sensitive modulation is observed at 532 nm and 405 nm within the experimental frequency range from 0.8 to 2.4 THz. The modulation depth reaches 80% under the 532 nm laser illumination with a maximum power of 250 mW and 96% under 405 nm illumination with a high power of 550 mW. The mechanism of the largely enhanced modulation depth is attributed to the construction of a type-II Bi_2_Te_3_/Si heterostructure, which could promote photogenerated electron and hole separation and increase carrier density dramatically. This work proves that a high photon energy laser can also achieve high-efficiency modulation based on the Bi_2_Te_3_/Si heterostructure, and the UV-Visible control laser may be more suitable for designing advanced all-optical THz modulators with micro-level sizes.

## 1. Introduction

Emerging terahertz (THz) technology has attracted considerable attention due to its unique applications in various fields, such as non-invasive detection [1], imaging [2], sensing [3], and wireless communication [4]. In particular, THz technology plays a significant role in the sixth-generation (6G) system with respect to meeting the unprecedented requirements for high-speed and large bandwidth in wireless communication [5]. However, there is still a huge bottleneck concerning developing high-efficiency and low-cost THz components, including THz sources, detectors, modulators, etc [6]. In particular, the THz modulator is an essential component in communication applications with respect to manipulating THz waves with a certain high frequency. The reported control methods include electrical [7,8], optical [9,10], thermal [11], and mechanical [12] modulation. Among them, electrical and optical modulation receive the most attention from researchers because the former is the best match to the current electrical systems, and the latter is the most suitable for future all-optical systems. Compared with the electrically driven THz modulators whose modulation speed is limited by the resistance-capacitance time constant [13], all-optical motivated THz modulators possess carrier lifetime determined modulation speed, as well as reduced power consumption and minimized cross-talk [6], which makes them available for certain attractive applications such as signal processing and optical communication systems [14].

The modulation performance of optical THz modulators is determined by the optical and electrical properties of the material, such as carrier mobility, carrier lifetime, and absorption coefficient [15]. In recent years, two-dimensional (2D) materials have prompted tremendous interest in designing novel optical THz modulators due to their unique optical and electrical properties, such as high carrier mobility, tunable carrier concentration, and various bandgaps [16,17]. However, the atomical thickness of 2D material limits the optical–matter interaction strength [16], hindering their modulation performance as the core material alone. One solution is to bond them with THz metasurfaces, which can improve the modulation depth greatly at resonance frequencies [18,19]. However, this method is generally locked in several narrow frequency regions. The other widely spread method is constructing heterostructure with classical photoresponsive semiconductor materials such as Si [20] and Ge [21], which has been proven to be effective in improving the modulation performance dramatically. From the start of the graphene/Si all-optical THz modulator being verified in 2012 [9], the majority of 2D material-based heterostructures, such as WS_2_/Si [22,23], MoTe_2_/Si [24], PtSe_2_/Si [25], and MoS_2_/Si [26], have been reported in the past seven years. The modulation depth is increased from 70% to 99.9% under the continuous wave (CW) laser illumination at several frequently used wavelengths of ~800 nm, ~1064 nm, and ~450 nm. However, the role of these different illumination wavelengths in the modulation performance has mostly been ignored. Note that the wavelength-dependent penetration depth of the control light has a significant change in the Si layer, which determines the device size. Understanding of the control wavelength-determined modulation mechanism and the advantages of short wavelength control light is essential for the development of micro-based compact all-optical THz modulators.

Bi_2_Te_3_ is a V–VI compound topological material with unique electrical and optical properties, including high carrier mobility, tunable bandgap, high conductivity, and single Dirac cone [27,28,29]. Recently, it has received enormous attention in photoelectric applications such as ultrafast laser photonics, optical communications, infrared sensors, and biomedical imaging [30,31,32]. In the THz region, 2D Bi_2_Te_3_ has proven to be a promising candidate for broadband photodetection [33], chiral THz emitter [34], and high-speed spintronic devices [35]. More importantly, exploring all-optical THz modulators based on Bi_2_Te_3_ is significant for expanding the THz application of this material, but relevant research has not been reported.

Herein, an all-optical controlled THz modulator based on Bi_2_Te_3_/Si heterostructure is investigated under 532 nm and 405 nm CW laser illumination. The Bi_2_Te_3_ film is prepared via the physical vapor deposition (PVD) method. The heterostructure achieves an adequate broadband modulation to the THz wave ranging from 0.8 THz to 2.4 THz. With the same laser power of 250 mW, the modulation depth of Bi_2_Te_3_/Si reaches almost the same values of ~80% and ~76% under 532 nm and 405 nm excitation, respectively. When further increasing the power to 550 mW at 405 nm, the modulation depth can reach as high as ~96%. The largely enhanced modulation performance is ascribed to the improved photogenerated electron and hole separation as well as the carrier lifetime and density driven by the type-II band alignment of the Bi_2_Te_3_/Si heterostructure. This work demonstrates the excellent THz modulation performance of the Bi_2_Te_3_/Si heterostructure controlled via UV-Visible light, which paves a promising route for designing novel all-optical micro-functional devices.

### Characterization of Sample and Experiment Setup

The Bi_2_Te_3_ films are prepared via the PVD method on sapphire and Si (>2000 Ω) substrates according to our previous works [36,37]. The schematic of the growth process is illustrated in Figure 1a. A total of 5 mg Bi_2_Te_3_ powder (99.99%, Alfa Aesar) as a precursor is put in the middle of the quartz tube, which is located in the heating zone of the furnace. Several sapphire and Si substrates are placed in a quartz boat behind the precursor at a distance of ~15 cm. Then, the temperature zone is heated to 550 °C with a rising rate of 15 min and maintained at the target temperature for 5 min. During the whole process, the atmosphere inside the quartz tube is protected by the Ar flow with a rate of 20 sccm.

Raman spectroscopy (SmartRaman confocal-micro-Raman module, Institute of Semiconductors, Chinese Academy of Science) of a Bi_2_Te_3_ film measured under a 532 nm CW laser excitation is presented in Figure 1b. The peaks located at 118.5 cm^−1^ and 134 cm^−1^ correspond to the infrared active mode A_1u_ and the Raman active mode $A_{1g}^{2}$, respectively [29]. The X-ray diffraction spectroscopy (XRD, Bruker, D8 Advance) of the Bi_2_Te_3_ film is shown in Figure 1c. There are four diffraction peaks representing different crystalline directions, which are indexed to hexagonal Bi_2_Te_3_ [38]. The scan electron microscope (SEM) image (Thermo Scientific, Apreo S., Waltham, MA, USA) of the Bi_2_Te_3_ film is presented in Figure 1d, and the inset shows the optical image (~1 × 1 cm^2^). These morphological characterization results on different scales prove that the film is uniform on a large scale. As shown in Figure 1e, the band gap of Bi_2_Te_3_ film is determined in the Tauc plot derived from the UV-Visible absorption spectroscopy.

The THz modulation performances of the samples are investigated via THz time-domain spectroscopy, which is illustrated in Figure 2a [39]. The THz wave is generated by exciting the ZnTe crystal with an 800 nm femtosecond laser. The THz wave is collected via a pair of off-axis parabolic mirrors, focused together with the probe laser on the detection crystal ZnTe and detected via the electro-optical sampling [40]. High-resistance Si (HR-Si) is utilized to block the femtosecond laser from the collinear THz wave. The 405 nm (or 532 nm) CW laser is then introduced with a normal incidence angle to modulate the THz transmission properties of the samples. Note that the spot diameter of CW light is about 4 mm, while the spot diameter of THz wave is approximately 3 mm. The experiment is performed with the protection of dry nitrogen. Figure 2b illustrates the transmitted THz wave modulation of a Bi_2_Te_3_/Si heterostructure driven by a CW laser which is geometrically expanded in the beam diameter to cover all the THz wave spots.

## 2. Results and Discussion

Figure 3a,b show the time-domain signals of the transmitted THz waves through Bi_2_Te_3_/Si and Bi_2_Te_3_/sapphire under 532 nm CW laser illumination with different power, respectively. Note that bare sapphire is transparent in the THz region, so the Bi_2_Te_3_/sapphire sample only reflects the effect of Bi_2_Te_3_, which exhibits negligible modulation to the THz amplitude with different illumination powers, as shown in Figure 3b. This result suggests that the photocarriers generated by topological Bi_2_Te_3_ itself under 532 nm CW illumination could not modulate the THz amplitude. However, for the Bi_2_Te_3_/Si heterostructure, the transmitted THz amplitudes decrease obviously on increasing illumination power, as shown in Figure 3a. This result indicates that the CW laser illumination could generate sufficient photocarriers in Bi_2_Te_3_/Si heterostructure to influence the transmission of THz waves.

Similar THz wave-modulation experiments have been performed on the same Bi_2_Te_3_/Si and Bi_2_Te_3_/sapphire samples under the illumination of a CW laser in the violet region of 405 nm, as shown in Figure 3c and 3d, respectively. The results show that the Bi_2_Te_3_/sapphire still has negligible modulation to the THz wave, but the Bi_2_Te_3_ samples can absorb the THz greatly when increasing the illumination power. For the Bi_2_Te_3_/Si, the decline in the transmitted THz amplitudes controlled via a 405 nm laser is more significant than that via a 532 nm laser. However, this enhancement may be induced by the higher maximum power of the 405 nm laser used in the experiment, as the THz transmission under the same laser power of 532 nm and 405 nm excitation is close. Compared with Bi_2_Te_3_/sapphire and bare Si [39], Bi_2_Te_3_/Si heterostructure has a significantly improved modulation capability with respect to the THz wave. This enhancement indicates that the heterostructure has a largely optimized modulation mechanism, which is discussed in the last part of the paper.

To further evaluate the modulation performance of the heterostructure, the frequency-domain THz amplitude spectra of Bi_2_Te_3_/Si under 532 and 405 nm CW laser excitation with different powers are obtained via Fourier transformation, as shown in Figure 4a and 4b, respectively. The THz wave generated and detected via ZnTe crystals in this work exhibits a broadband feature that covers the 0.4–2.8 THz range. The high signal-to-noise ratio region is around 0.8–2.4 THz, and the two peaks are at ~1.4 and ~2.1 THz. Therefore, the evaluated frequency range is wider than the reported THz modulation works based on heterostructures such as WS_2_/Si (0.2–1.6 THz) [23] and PtSe_2_/Si (0.1–1.5 THz) [25]. As a result, the modulation of the Bi_2_Te_3_/Si heterostructure is broadband-effective and covers the whole experimental frequency region. Moreover, the THz amplitude decreases gradually with the increase in illumination power under both 532 nm and 405 nm CW laser excitation. Here, the modulation effect driven by the 405 nm laser is greater than that by the 532 nm laser. 

The modulation depth (M) that represents the ability to modulate the THz intensity of the device can be calculated quantitatively as $M=\frac{A_{\mathrm{off}}-A_{\mathrm{on}}}{A_{\mathrm{off}}}\times 100\%$. Here, *A*_off_ and *A*_on_ are the THz peak-valley values without and with the CW laser illumination. Under the 532 nm and 405 nm CW laser illumination with different powers, the modulation depth of Bi_2_Te_3_/Si in the frequency domain is presented in Figure 4c and 4d, respectively. The heterostructure exhibits a broadband modulation feature in all the conditions as mentioned above. Taking the 1.4 THz position as an example, the modulation depth under 532 nm excitation increases gradually from 25% to 80% when increasing the illumination power from 50 to 250 mW, as shown in Figure 4e. The modulation depth increases with the enhancement of illumination power in the experimental measurement range. In comparison, under 405 nm excitation, the modulation depth of Bi_2_Te_3_/Si rises from 7% to 94% with the enhancement of illumination power from 50 to 550 mW (Figure 4f). Note that in the relatively low illumination power region (<250 mW) with the same photon number, the modulation depth of Bi_2_Te_3_/Si under 532 nm (250 mW) excitation is 76%, which is superior to the modulation depth of ~49% under 405 nm (190 mW) excitation. However, the modulation depth grows faster in the low illumination power region for the 405 nm excitation condition and becomes more significant in the higher illumination power region up to 550 mW. As a result, both the 405 nm and the 532 nm CW lasers for Bi_2_Te_3_/Si show excellent broadband modulation performance in the THz region. Within our experimental conditions, a 405 nm ultraviolet laser at high power shows better THz amplitude modulation ability than 532 nm visible light.

To elucidate the modulation mechanism, the modulation response properties of the Bi_2_Te_3_/Si heterostructure are assessed through the examination of its illumination-on/off behavior. Figure 5a shows the time response of the transmitted THz amplitude controlled via the 532 nm CW laser with a power of 250 mW. This result suggests a clear THz amplitude modulation from 0 to 74% under illumination-off to -on conditions. The photoresponsivity (R) is generally described in units of THz absorption per watt of incident laser power [41,42]. Hence, the R is defined by the analytical equation $R=\frac{A_{\mathrm{off}}-A_{\mathrm{on}}}{P}$, where *P* is the illumination power. The photoresponsivity of the Bi_2_Te_3_/Si heterostructure is calculated to be ~1.76 W^−1^. Similarly, the time series of illumination-off and illumination-on for THz amplitude of Bi_2_Te_3_/Si is presented in Figure 5b under 405 nm CW laser illumination with the same power. Note that the THz signal is almost completely shielded under the illumination-on condition, and the photoresponsivity is calculated to be 2.4 W^−1^, which is ~1.4 times larger than that of the 532 nm CW laser excitation condition. This indicates that the THz modulation performance is superior for the 405 nm CW laser illumination condition.

Additionally, the response and recover time under both 532 and 405 nm laser excitation is approximately 1 s, as shown in Figure 5. However, this long time parameter is dominated by the mechanical baffle switching time which is close to 1 s instead of the real modulation speed governed by the carrier dynamic process of the sample. Therefore, one can expect a much faster modulation speed in the future when using a Bi_2_Te_3_/Si THz modulator controlled via choppers or other electrical control units.

Next, the modulation mechanism of the Bi_2_Te_3_/Si heterostructure under CW laser illumination is discussed. The THz amplitude transmission T(ω) of the Bi_2_Te_3_/Si heterostructure is related to its THz photoconductivity σ(ω) according to Tinkham’s equation [43]:(1)$$\sigma (\omega )=\frac{\left(1+n_{s}\right)}{Z_{0}d}\left(\frac{1}{T(\omega )}-1\right)$$
where $n_{s}$ is the refractive index of Si, and *Z*_0_ and *d* are vacuum impedance and photoconductive layer thickness, respectively. In the THz region, complex photoconductivity of semiconductor materials generally follows the Drude model [44]:(2)$$\tilde{\sigma }(\omega )=\frac{\omega_{p}{}^{2}\varepsilon_{0}\tau }{1+i\omega \tau }$$
where $\omega_{p}$, $\varepsilon_{0}$, τ, and ω are plasma frequency, vacuum permittivity, lifetime of photogenerated carriers, and THz frequency, respectively. Further, the plasma frequency is related to the carrier density *n* as $\omega_{p}{}^{2}=ne^{2}/m^{*}\varepsilon_{0}$, where *e* and *m^*^* are electron charge and effective electron mass, respectively. Therefore, the THz transmission is related to the photoconductivity and the carrier density, which is largely affected by the material’s optoelectronic properties and the illumination laser conditions. In a simplified intrinsic absorption model, the relationship between the average photogenerated charge carriers in the photoconductive layer and the carrier lifetime can be estimated with the following formula [25,45]:(3)$$n=\frac{PA\alpha \tau }{h\nu Sd}$$
where *A* is the absorptivity (excludes the reflection and transmission losses), *α* is the responsivity that represents the photocarrier-generation efficiency, *hν* is the photon energy, and *d* is the layer thickness. Note that the photogenerated charge carriers are not uniformly distributed in real semiconductors, but this model can still reflect the influence of the key factors. For example, according to Equations (2) and (3), the carrier density is proportional to the photoconductivity, and to the illumination power. According to Equation (1), the carrier density is inversely proportional to the THz amplitude transmission with an offset, which is proportional to the opposite amont of modulation depth. Therefore, as shown in Figure 4e,f, the THz modulation depth is enhanced by increasing the laser power.

To elucidate the charge transformation mechanism of the Bi_2_Te_3_/Si heterostructure under CW laser excitation, the carrier separation process at the interface is discussed in the following part. The energy band alignment of the Bi_2_Te_3_/Si heterostructure before and after contact are shown in Figure 6a and 6b, respectively. Here, E_vac_ is the vacuum level, and Ф_1_, $\chi_{1}$, E_c1_, E_v1_, E_F1_, and E_g1_ are work function and electron affinity, conduction band bottom, valence band maximum, Fermi level, and band gap of Bi_2_Te_3_, respectively. Similarly, Ф_2_, $\chi_{2}$, E_c2_, E_v2_, E_F2_, and E_g2_ are the corresponding parameters of Si. ΔE_c_ and ΔE_v_ are defined to be the difference between the two conduction bands and valence bands, respectively. In this experiment, the Bi_2_Te_3_ film prepared via the PVD method is the same as the material reported in our previous work for photoelectrochemical applications, so the band alignment parameters can be found in the UV-Vis absorption spectroscopy, X-ray photoelectron spectroscopy, and ultraviolet photoelectron spectroscopy results [36]. E_c1_, E_v1_, E_F1_, and E_g1_ are about 5.02 eV, 5.89 eV, 5.86 eV, and 0.87 eV, respectively. The Si substrate is slightly *n*-doped and we estimate the band alignment with parameters according to Meng’s report [46], in which E_c2_, E_v2_, E_F2_, and E_g2_ are around 4.29 eV, 5.41 eV, 4.69 eV, and 1.12 eV, respectively. Therefore, ΔE_c_ and ΔE_v_ are calculated to be ~0.73 eV and ~0.48 eV, respectively. After the contact between Bi_2_Te_3_ and Si, the Fermi levels reach the same value for an equilibrium state and a type-II heterostructure is formed as illustrated in Figure 6b. When the CW laser illuminates the sample, an interband transition occurs in both Bi_2_Te_3_ and Si. Under the built-in electric field, photogenerated electrons transfer from Bi_2_Te_3_ to Si while holes are in the opposite direction, as shown in Figure 6c. The separation of electrons and holes can be largely enhanced by the band alignment of the type-II heterostructure. The separation of electrons and holes significantly reduces the recombination of photocarriers, leading to an enhancement of the carrier lifetime as well as the carrier density, so the THz modulation depth is largely improved by the more sensitive photoconductivity.

In this work, the photon energies of the 405 nm and 532 nm lasers are larger than the band gap energies of both Bi_2_Te_3_ and Si. Therefore, the THz transmission properties of single Bi_2_Te_3_ or Si can be explained via the model mentioned above. For the Bi_2_Te_3_/sapphire sample, Bi_2_Te_3_ is the only photoconductive layer because the absorptivity of the sapphire substrate is zero. However, the absorptivity of Bi_2_Te_3_ is also limited by the very thin film thickness and the lifetime of photogenerated carriers is short due to electron–hole recombination. As a result, the photocarrier density in Bi_2_Te_3_ is very low and thus the modulation is almost negligible, as shown in Figure 3b,d. For bare Si, the absorptivity under both 405 nm and 532 nm excitation is high (only the reflection loss should be considered) because the film thickness is far longer than the penetration depth, which can be regarded as the thickness of the photoconductive layer. Hence, the modulation of bare Si can be observed in many optically controlled THz modulation works [22,39,47,48]. However, the modulation depth of bare Si is low because the recombination of photocarriers limits the lifetime and carrier density. Therefore, the effective separation of electrons and holes, which can be realized via reasonable heterostructure construction, becomes an important way of designing high-performance THz modulators.

Lastly, the modulation performances of 532 nm and 405 nm CW lasers are compared and discussed. Because both of the two wavelengths can induce interband transition in the Bi_2_Te_3_/Si heterostructure, the main difference could originate from the factors given in Equations (1–3). Under 532 nm and 405 nm CW laser excitation, the penetration depths of Si are ~1 and ~0.1 µm, respectively [49,50]. With the increase in illumination power, the photoconductivity and carrier concentration exhibit a growth trend in both the wavelength cases. As shown in Figure 7a,b, when the power is enhanced from 0 mW to 250 mW, the conductivity changes from 3.69 × 10^3^ S/m to 3.25 × 10^4^ S/m, and the carrier density varies from 7.18 × 10^13^ cm^−3^ to 6.34 × 10^14^ cm^−3^ under 532 nm excitation. In contrast to the previous statement, when exposed to 250 mW illumination power under 405 nm excitation, the photoconductivity and carrier density can reach 6.13 × 10^5^ S/m and 1.19 × 10^16^ cm^−3^ (Figure 7c and d), respectively. This enhancement is mainly induced by the narrow penetration depth at 405 nm, but the modulation performance under 405 nm and 532 nm excitation has no significant difference because the *d* factor is eliminated in Equation (1). When the illumination power further increases to 550 mW, the conductivity and carrier density of the photoconductive layer can reach 5.84 × 10^6^ S/m and 1.14 × 10^17^ cm^−3^, respectively. Note that the laser power dependence of the carrier density measured in the experiment is not linear as predicted in Equation (3), especially in the high-power region. This could be attributed the enhancement of the carrier lifetime induced by the heterostructure, which is ignored in the calculation. These results may provide a meaningful reference from the perspective of microdevice design. According to previous reports [49,50], the penetration depth values of Si for several typical laser wavelengths of 405 nm, 532 nm, 800 nm, and 1064 nm are ~0.1 μm, ~1 μm, ~10 μm, and ~900 μm, respectively. This shows that the penetration depth changes dramatically with the excitation wavelength. Therefore, the laser with higher photon energy may make sufficient utilization of the carrier separation capacity at the finite heterostructure region. In addition, the modulation devices could be fabricated smaller and more compactly when controlled via short-wavelength lasers. For example, Si films with submicron thicknesses could be sufficient for the THz modulator design under 405 nm excitation, which can also achieve a modulation depth close to 100% as verified in this work.

## 3. Conclusions

An all-optical THz modulator based on Bi_2_Te_3_/Si heterostructure is investigated in this work. The 2D Bi_2_Te_3_ is prepared via the PVD method on sapphire and Si substrates directly, and these samples are measured via THz time-domain spectroscopy. Under the control of CW lasers at 532 nm and 405 nm, broadband modulation performance is observed in the 0.8–2.4 THz region. The modulation depth can reach ~80% under 532 nm CW laser illumination with a power of 250 mW, and ~96% under 405 nm illumination with a higher power of 550 mW. The significantly enhanced modulation capacity of the Bi_2_Te_3_/Si compared to that of Bi_2_Te_3_ is attributed to the highly improved photocarrier separation at the type-II heterostructure interface, and the modulation mechanism is discussed with the help of the Drude model and a simplified light absorption model. This work provides a useful reference for the design of compact all-optical THz modulators with no loss of performance.

## Figures and Tables

**Figure 1 micromachines-14-01237-f001:**
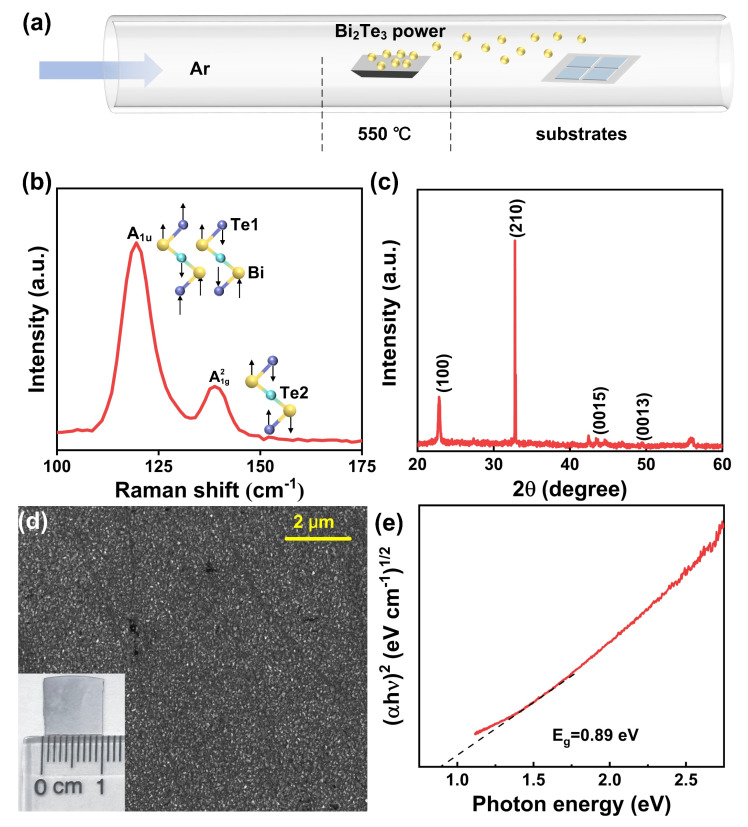
(**a**) Schematic of the PVD growth process of Bi_2_Te_3_ films on different substrates. (**b**) Raman spectrum of the Bi_2_Te_3_ film under a 532 nm laser excitation. The inset is the schematic of vibration modes. (**c**) XRD of the Bi_2_Te_3_ film. (**d**) SEM image of the Bi_2_Te_3_ film. The inset is the optical image. (**e**) Tauc plots ((α*h*ν)^1/2^ versus *h*ν) give the band gap of Bi_2_Te_3_.The red line and black dotted line represent the absorption edge and tangent line, respectively.

**Figure 2 micromachines-14-01237-f002:**
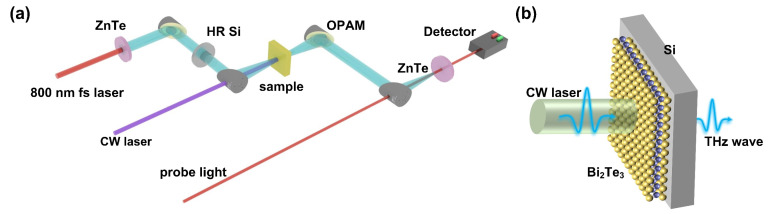
(**a**) Schematic of the THz time-domain spectroscopy system for the CW laser-controlled modulation experiment. (**b**) Illustration of the Bi_2_Te_3_/Si heterostructure THz wave modulation via CW laser.

**Figure 3 micromachines-14-01237-f003:**
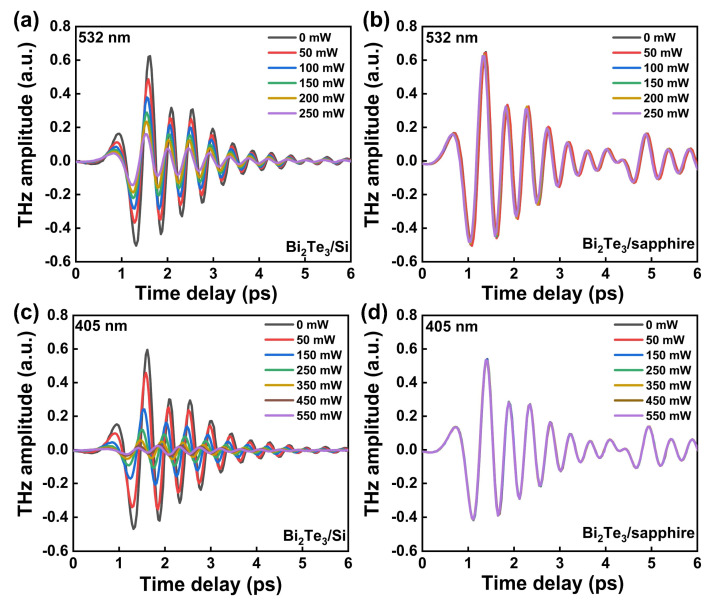
Time-domain signals of the transmitted THz waves through (**a**) Bi_2_Te_3_/Si and (**b**) Bi_2_Te_3_/sapphire under 532 nm CW laser excitation with different pump powers. Time-domain signals of the transmitted THz waves through (**c**) Bi_2_Te_3_/Si and (**d**) Bi_2_Te_3_/sapphire under 405 nm CW laser excitation with different pump powers.

**Figure 4 micromachines-14-01237-f004:**
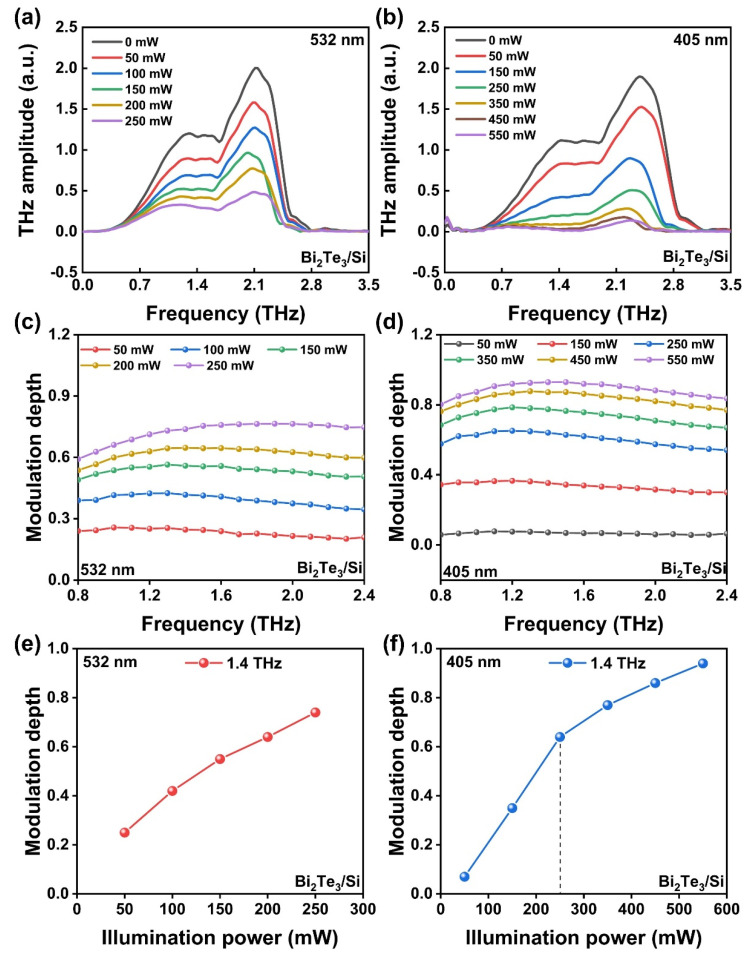
Frequency-domain THz amplitude spectra of Bi_2_Te_3_/Si under (**a**) 532 nm and (**b**) 405 nm CW laser excitation with different pump power. Modulation depth of Bi_2_Te_3_/Si under (**c**) 532 nm and (**d**) 405 nm CW laser excitation with different pump power. Illumination power-dependent modulation depth of Bi_2_Te_3_/Si at 1.4 THz under (**e**) 532 nm and (**f**) 405 nm CW laser excitation.

**Figure 5 micromachines-14-01237-f005:**
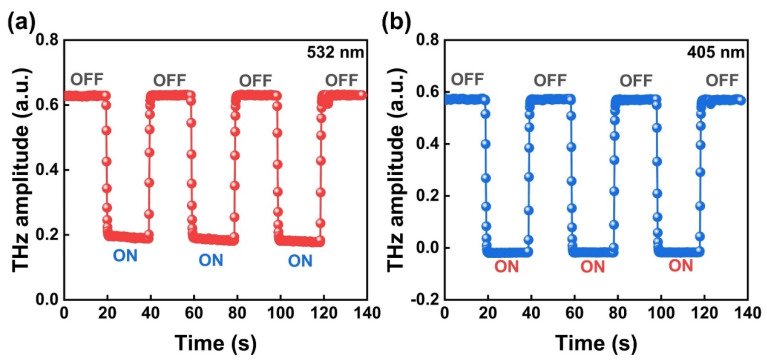
Time series of illumination-off and illumination-on about the THz transmission amplitude under (**a**) 532 nm and (**b**) 405 nm CW laser excitation with 250 mW.

**Figure 6 micromachines-14-01237-f006:**
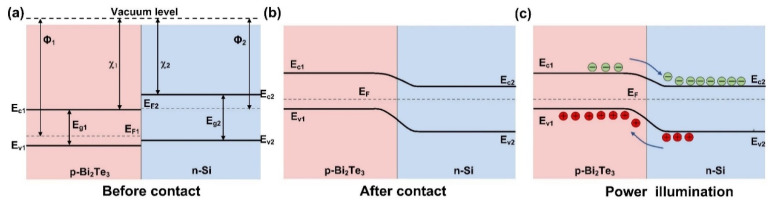
(**a**) Energy band structure of Bi_2_Te_3_ and Si before contact. (**b**) Band alignment of the Bi_2_Te_3_/Si heterostructure after contact. (**c**) Band alignment of the Bi_2_Te_3_/Si heterostructure under laser illumination.

**Figure 7 micromachines-14-01237-f007:**
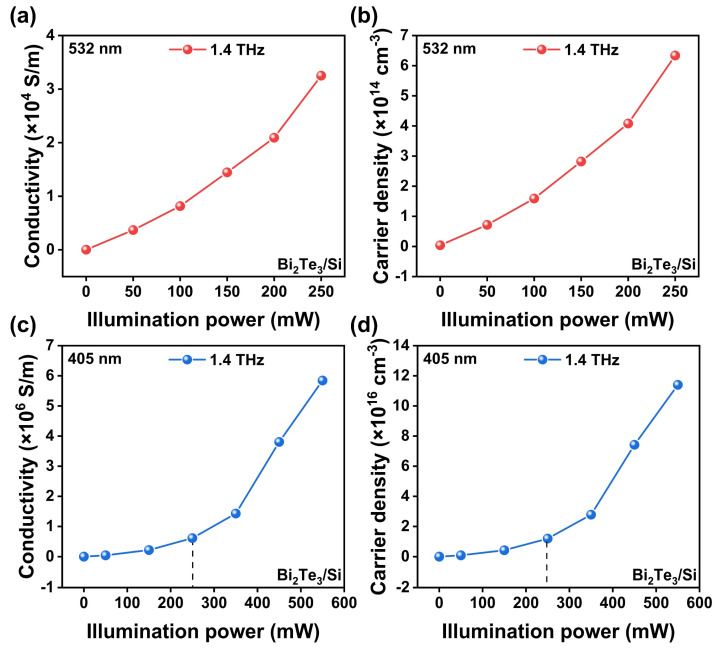
(**a**) Photoconductivity and (**b**) carrier concentration of Bi_2_Te_3_/Si photoconductive layer under 532 nm excitation with different illumination powers. (**c**) Photoconductivity and (**d**) carrier concentration of Bi_2_Te_3_/Si photoconductive layer under 405 nm excitation with different illumination powers.

## Data Availability

Not applicable.

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
