# Peer review of "Broadband All-Optical THz Modulator Based on Bi2Te3/Si Heterostructure Driven by UV-Visible Light"

_micromachines, 2023, doi:10.3390/mi14061237_

Round 1

Reviewer 1 Report

The manuscript by Xi et al. studies an all-optically terahertz (THz) modulator based on Bi2Te3/Si heterostructure under UV-Visible light excitation by the THz time-domain spectroscopy. The modulation depth can be up to ~80% for the 532 nm CW laser illumination with 250 mW power and ~96% for the 405 nm illumination with a higher power of 550 mW. The enhanced modulation performance is attributed to the highly improved photocarrier separation at the type-II heterostructure, and the mechanism is discussed with the help of the Drude model and a simplified photo absorption model. The experimental results are solid and the topic is suitable for Micromachines. I should be glad to recommend this manuscript for publication if the authors would address the following comments:

1.     Does the sample absorb equally under 532 nm and 405 nm CW laser excitation?

2.     I wonder why the sample was excited at 532 nm with 250 mW instead of 550 mW?

3.     Is the spot area of the continuum light the same as that of the THz wave? Please include experiment parameters in this manuscript.

4.     Why is the photoconductivity and carrier concentration of Bi2Te3/Si calculated at 1.4 THz?

5.     While the overall writing of the manuscript is clear, the author should be careful with some of the wording. Also, check with a native speaker or some software to improve the writing.

None

Reviewer 2 Report

This paper reports an all-optical controlled THz modulator based on Bi2Te3/Si heterostructure. It is a new application of Bi2Te3/Si to the field of THz modulation. Good modulation capability has been shown up to 96% under 405 nm laser excitation, showing that Bi2Te3/Si is also a promising candidate for the THz modulator. Researchers of related fields will be interested in this work. I suggest this work be published after proper and careful revision with the following questions being answered appropriately.

1.      Fig.3(b) shows slight time delay between waveforms of various illumination power, whereas Fig.3(d) does not. Does it come from the errors of repeated measurements? If not, please give an explanation.

2.      The laser / THz beam diameters on the sample are important in this work. Please supply the information when introducing your experimental setup.

3.      I suggest that Fig.4 (c) and (d) use the same vertical range so that readers can easily compare them.

4.      The modulation speed is very important for modulators designed for communications. The authors made On-Off test of their sample. Please supply more information of the transition rate. From Fig. 5, it seems the switch speed is too low to meet the demands. Please make some discussions and suggestions of improving the performance in the future.

5.      The authors define the modulation depth at line 174, in which the THz amplitude is used. But, according to the statement at line 173, THz intensity (or energy, proportional to the square of the THz amplitude) should be used to obtain the modulation depth. According to Eqs. (1) to (3), the carrier density is proportional to the photoconductivity, and to the illumination power. The carrier density is inversely proportional to the THz transmission with an offset. Since in this paper, the modulation depth is calculated using the amplitude of transmitted THz wave (the square root of the transmitted THz power). So, the modulation depth is inversely proportional to the square root of the THz transmission under laser illumination. Then, we can see that in Fig.4, the curves follow the square root function. That means the authors made wrong discussions in many parts of the paper, like “The modulation depth shows a local linear dependence on the illumination power in the experimental measurement range.” (Line 181). And, the authors should explain why the results of the carrier density and the photoconductivity seem to be the square of the illumination power. The authors should make clear explanations to prove that their results and discussions are reliable.

6.      The authors define the photo responsivity in line 203. It is inversely proportional to the product of the illumination power and the area. Please explain how they get this equation. I don’t think it is reasonable to calculate the photo responsivity by multiplying the illumination power (not power density) and the area.

7.      The author compared the modulation depth in the same power region (below 250 mW) under 532 nm and 405 nm illumination at lines 185-187. I don’t think it is proper to do so since the photon energy is different. The authors should compare the results with the same photons, not the same power. Because the transmission of THz is related to the photogenerated carrier density, which is directly related to the incident photons. So, the authors should revise their discussions to give proper analysis.

8.      The author should pay more attention to the editing. For example, the “power” in line 91 and Fig.1(a) should be “Powder”. In Eq. (2), is t “the photogenerated carriers scattering rate” or “the lifetime of carriers”?

Generally, the English language is good, except for some mistyping. Some sentences are a little weird for me, feeling like they were translated by machine.

Reviewer 3 Report

This paper reports a detailed experimental investigation of the potential of the topological material Bi2Te3 and Si heterostructure as an all-optical terahertz (THz) modulator and is worthy of publication in this special issue. However, the following needs to be addressed.

 p.8  The time response of the modulation is studied in Figure 5, but the response time of the modulation is not clearly shown. If possible, the response function should be clarified by driving the sample with impulse illumination. At least, the response time should be estimated from the present results. 

The following related papers should be cited.

 Wang, X.B., Cheng, L., Wu, Y. et al. Topological-insulator-based terahertz modulator. Sci Rep 7, 13486 (2017). https://doi.org/10.1038/s41598-017-13701-9

PHOTONIC SENSORS / Vol. 9, No. 3, 2019: 268‒276

Optically Controlled Extraordinary Terahertz Transmission of Bi2Se3 Film Modulator

Junhu ZHOU, Tong ZHOU, Dongsheng YANG, Zhenyu WANG, Zhen ZHANG, Jie YOU, Zhongjie XU, Xin ZHENG, and Xiang-ai CHENG

 Minor corrections:

p.2  Line 91  5 mg Bi2Te3 power   powder

p.4  L143  in the ultraviolet region of 405 nm      violet
